

# Comparison of machine learning classification algorithms for land cover change in a coastal area affected by the 2010 Earthquake and Tsunami in Chile.

Matias I. Volke[1], Rodrigo Abarca-Del-Rio [1]

[1] Department of Geophysics, Faculty of Physical and Mathematical Sciences, University of Concepcion, Concepcion, Chile.

*Correspondence to:* Matias I. Volke (matiasvolke@udec.cl)

**Abstract:** Earthquakes and tsunamis are the natural events that generate subsequent geomorphological land cover changes. The damage is usually of such importance and of such a diverse nature that it is imperative to have tools that allow quick and precise monitoring. Thus, know which classification methods have the best potential to obtain greater precision will improve natural disaster management. We analyze Tubul locality (37.21ºS; 73.43ºO) in Biobío region, Chile, in which greatest geomorphological changes were documented after the earthquake and tsunami occurred 27/February/2010. These changes can be analyzed using different machine learning methods. We investigate the "Support Vector Machine" (SVM) and "Random Forest" (RF), versus the "Maximum Likelihood" (ML) classification method of Landsat TM and ASTER satellite images. The comparison of the performance of the classifiers and certifying accuracy improvement shows that machine learning algorithms are superior to traditional classification methods in terms of overall accuracy and robustness. The general classification accuracy was approximately 97%. We also visualize the land cover transformations, showing that 26% of the region was altered. The results of performance testing of machine learning methodologies was consistent with other studies and presents a valid application in the visualization of land cover changes in areas of natural disasters.

## 1.-Introduction

Multispectral images provide an effective tool for various fields of research related to land cover and land-use change (Chuvieco, 2010; Maxwell et al., 2018a). Over the last decade, observations of cover change have enabled vulnerability studies in areas at risk or affected by natural disasters, with emphasis on populated areas, thanks to image comparisons, such as those used to study natural disasters like earthquakes and tsunamis (Ma et al., 2016; Satheesh Kumar et al., 2008; Wu et al., 2016).

Coastal risk mapping based on satellite images is an application of land cover change studies (Kaiser et al., 2013; Römer et al., 2012). Several studies have already used remote sensing to show areas affected by destructive natural phenomena such as tsunamis, highlighting those that showed the changes produced by the tsunami that affected Japan on the 11th March 2011. Richmond et al. 2012, reports on soil changes associated with the Tohoku-Oki tsunami by contrasting satellite images. Tappin et al. 2012, by interpreting satellite image time series documented coastal change. Ishihara and Tadono 2017, using Landsat 8 images, presented a time series of maps over the affected area of Tohoku. In addition, Kaiser et al 2012, conducted a spatial and temporal evaluation of the impacts of the Indian ocean tsunami, by remote sensing. Also Sarun et al. 2018 , focused their work on the analysis of the previous and subsequent Sumatra 2004 tsunami scenario, starting with 16 years of Landsat ETM + and OLI multispectral data.

Classification techniques are fundamental to get reliable information from satellite data. Over the past decades, several classification approaches have been developed (Dhodhi et al., 1999; Maxwell et al., 2018b). Within the literature, we can distinguish the traditional and advanced classifiers. Among the traditional classifiers stand out ISODATA, k-means and ML (Blanzieri and Melgani, 2008; Rawat and Kumar, 2015; Rogan et al., 2008). While in the advanced stand out; neural networks (ANN; (Bocco et al., 2007)), decision trees (DT ;(Pal and Mather, 2003)), random forest (RF;(Belgiu & Drăguţ, 2016) ) and support vector machine (SVM; (Mountrakis et al., 2011a)). However, the selection of the classifier, bands (original or derived) and parameter definition by the user are prerequisites to improve the accuracy of the classification (Maxwell et al., 2018b). Reviews such as Mountrakis, Im, and Ogole 2011a and Lawrence





and Moran 2015, present an overview of recent methodologies and applications of classification models in
remote sensing.

We propose a machine learning algorithm implementation such as SVM and RF, having as the main
motivation, improvement in performance, accuracy, and reliability over the classification results achieved
by traditional methods such as ML. Another source of motivation is that in the remote sensing related field
of coastal changes related to Tsunami, these classifiers (SVM and RF) are not as familiar as other classifiers,
such as decision trees (DT), and neural network (NN) variants.

We classified the satellite images involving the periods before and after February 27, 2010, concentrating
on the Tubul town area, in central Chile. On this date, an earthquake of magnitude Mw = 8.8 (according to
the Chilean seismological service) reached the central Chile coast with its epicenter 50 km northeast of the
Concepción city, at a depth of 47.7 km (see Figure 1). It caused a magnitude 4 tsunami, affecting small
bays in a coastal stretch of 800 km along the Maule and Biobío regions (Quezada et al. 2012.). The
earthquake and tsunami events generated destruction of the infrastructure of cities and coastal towns
(Cienfuegos et al., 2014; Soto et al., 2015) and profound modifications in various aspects in the territory's
geography, among which stand out coseismic uplift and subsidence, seawater entry in coastal areas, sand
and debris transport, erosion of dunes and coastal bars (Araya and Carvajal, 2016; Martínez et al., 2011).
Within all the coastal areas, one of the most affected, and for which, there are a remarkable number of
descriptive reports, is the Tubul area. In this town, tsunami resulted in a flooding area of about 75 ha., while
destructing the sector made up by the edge of the beach and parts of the banks of the river (Martínez et al.,
2011). These changes are those that we wish to visualize by implementing advanced classification methods
that will allow categorizing identifiable land cover changes.

## 2-Materials and method

### 2.1- Study area

We examined the Tubul location (37,21 °S; 73,43 °W) in the Arauco Province, BioBío Region (see Figure
1), south-central Chile. As pointed out in the introduction, the earthquake and following Tsunami of
February 27, 2010, affected the coast of the Arauco Bay, near the Tubul-Raqui wetland, which has two
main watercourses, Tubul and Raqui rivers. A sandbar of recent (quaternary) fluvial-marine sedimentation
forms a very low plain at the foot of a paleo-cliff and the fluvial Tubul-Raqui wetland system. Prior to the
tsunami, sand sediments dominated the coastal dynamics in the low plain, forming a bar, the human
settlement located on both the dunes and sands of the bar. These features provide only limited natural
protection from coastal inundation, and thus the areas are of known vulnerability to marine events. As a
result, the tsunami altered Tubul, though not destroyed, favored by its orientation, which is not open to the
north from where the wavefront arrived (Martínez et al., 2011).

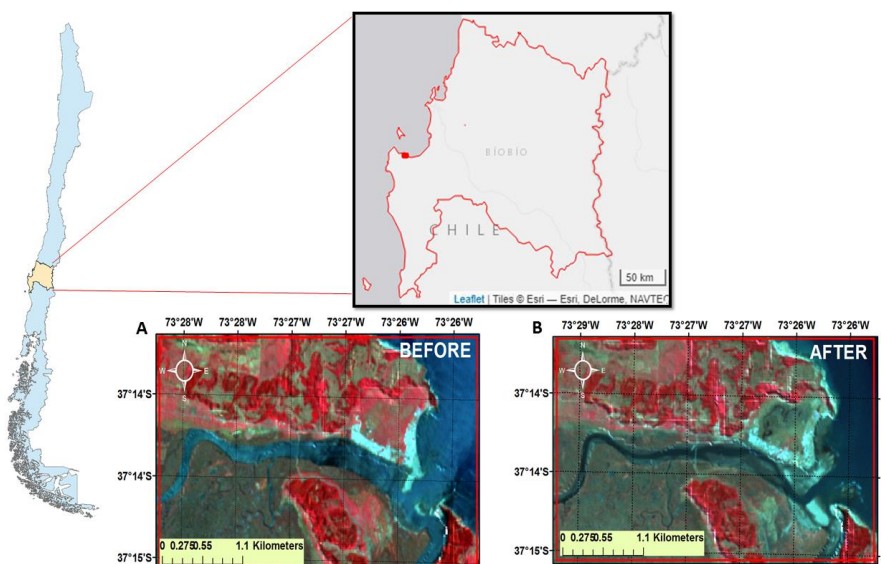

**Figure 1.** Tubul (37.21 °S; 73.43 °W) in Arauco province, Biobío region. A) RGB ASTER image (before 27/F);
B) RGB ASTER image (after 27/F).

### 2.2. - Multispectral Images of Moderate Resolution.

We chose pre- and post-earthquake geo-referenced Landsat and Aster satellite images that could likewise
be comparable, that is dates with comparable meteorological conditions. It is the main reason we resolved
not selecting images days previous to the event, but from the same time as the preceding year. In Table I,
we present the characteristics of these images. In addition, the free-access SRTM (Shuttle Radar
Topography Mission) digital elevation model (DEM) was used, with a spatial resolution of 30 meters,
corresponding to the area of study.

**Table 1.** Descriptive table of the satellite images used.

| Satelite | Year | DD / MM / YYYY | Description | Bands |
|----------|------|----------------|--------------------|-----------|
| Landsat  | 2009 | 07/03/2009     | path001 / row085   | 1-5,7     |
|          | 2010 | 26/03/2010     | path001 / row085   |           |
| Aster    | 2009 | 27/02/2009     | AST_L1T_0030227    | 1-3 (VNIR)|
|          | 2010 | 11/03/2010     | AST_L1T_0030311    |           |

### 2.3. - Data preparation

Prior to the analysis, the images must go through some processes to give a physical sense to the values,
with the methodology expressed in figure 2.

Different corrections are necessary: First, we make a radiometric correction to get the reflectance values at
the top of the atmosphere (TOA) according to the procedure described in (Chander et al., 2009). Because
of varying sun-ground-sensor geometry, which is affected by the topography of the area, we also perform
a topographic correction. This imposes an additional variation on the radiometric data in pixels with ground
cover and very similar structural biophysical characteristics, but with characteristics of terrain slope or


different zenith and azimuth angles at the time of capture (Moreira and Valeriano, 2014). The "correction C" detailed in (Soenen et al., 2005) was applied. Composition is made from the 6 bands extracted from Landsat and the 3 bands extracted from ASTER. The training and validation samples, which contain representative pixels for the different soil coverages, were extracted by selecting a region of interest (ROI). In ROI's a simple sampling selection protocol was followed to define the training samples. Later, it is

compiled in a general list with the values of the pixels for each band.

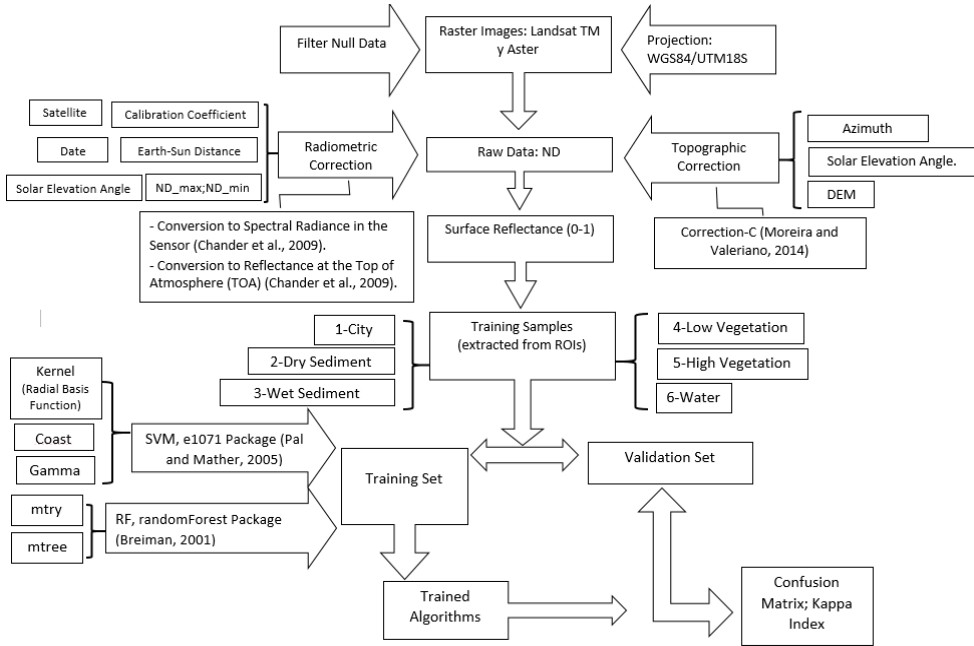

**Figure 2**. Methodological scheme to illustrate steps necessary for classification using the SVM and RF algorithms.

**Table 2**. Definition of the thematic classes that represent the soil cover selected to represent the reality of land.

| Soil cover | Description |
|---|---|
| 1. City | Urban soil and exposed rocks that, like cement, have a high albedo; It reflects a large amount of incident energy. |
| 2. Dry Sediment | Exposed soil with low moisture content and organic matter, such as sand, exposed hillsides, uncultivated areas, cleared areas, burned areas, erosion areas and areas with no vegetation. |
| 3. Wet Sediment | Soils with high moisture content, such as wetland soils. Also, farmland and coastal deposits. |
| 4. Low vegetation | Plant formation where the herbaceous cover is over 40%, this includes land with crop rotation, types of trees and shrubs with an area of extension of less than 25%. Areas used by agriculture, including cereal crops, vegetables, and fruits. |
| 5. High Vegetation | Vegetation cover in which the tree stratum is established by natural species such as Coihue, Olivillo, Patagua, and Boldo. It also represents forests where the arboreal stratum is formed by exotic species such as eucalyptus and radiata pine. |
| 6. Water | Surfaces covered by water, both fresh and salty. |



### 2.4. - Classification of images

We executed the classifications of the images using ML, RF, and SVM. In the following subsections, we present a brief description of the 3 algorithms considered.

### 2.4.1 - Maximum Likelihood

The Maximum Likelihood (ML) method is one of the most used classification methods in remote sensing (Yonezawa, 2007). This method is based on the assumption that the image DN (digital number) values in each of the user-defined classes follow a multivariate normal probability distribution. Although this assumption is not always true, the method is robust (Ahmad & Quegan, 2012).

### 2.4.2. - Random Forest

RF is a learning algorithm that combines information from an ensemble of decision trees using random subsets of variables to classify and train data (Breiman, 2001; He et al., 2017). The trees vote to determine the label assigned to unknown samples. This overcomes the problem that any tree is non-optimal, as when incorporating many trees, it takes a global optimum (Rodriguez-Galiano et al., 2012). The set of decision trees or "forest" is built up from the training data selected by the user executing a "bootstrap" sampling. In
this sampling, only 2/3 of the original training data for each tree are randomly used, and besides a random selection of predictor variables is used to split the nodes in the tree's construction (Naidoo et al., 2012).

As detailed in Dye et al. (2011) and He et al. (2017), there are two main adjustment parameters required to configure an RF algorithm. These parameters are the number of trees that will be built in the forest and the number of division categories considered for each node in the trees. RF uses an out-of-bag (OOB) procedure
where the remaining 1/3 of the training samples (randomly picked out and taken out from the sample to establish the decision tree) are booked as an internal test set, which calculates so an unbiased and reliable error rate (Maxwell et al., 2018b).

### 2.4.3 - Support Vector Machine.

SVM focuses only on the training samples closest to the space characteristics to the optimum limit of
separation between the classes (Vapnik, 1995). These samples are called support vectors and are used to define the hyperplane with the maximum margin (i.e., separation) between classes (Mountrakis et al., 2011a).

The basic linear decision limits are often not enough to classify the categories with high precision. Techniques and alternative solutions such as a kernel function used to work out the problem of
inseparability, introducing additional variables in the optimization of SVM and mapping (through an adequate mathematical function) the non-linear correlations in a higher space (Euclidean or Hilbert) . The selection of a kernel function often influences the results of the analysis, so in the same way as the adjustment parameters, it is very important to choose it carefully (Kavzoglu and Colkesen, 2009). In addition, SVM requires the "cost " parameter (C parameter) definition, which controls the cost paid by the
SVM for erroneous classifications of a training point. Adjustment of this parameter can balance the margin maximization and the classification violation (Melgani and Bruzzone, 2004). Interestingly, SVM does not assume a known statistical distribution of the data to be classified. This is useful since data gained from remote sensing images have an unknown distribution and normality does not always provide a correct assumption of the actual dispersion of the pixels in each class.

### 2.4.4. - Validation of algorithms

We created a confusion matrix for global accuracy and analysis of the reliability of the implemented models (Tralli et al., 2005). We calculated the kappa statistic for algorithm evaluation, which tests the success of pairs of data between a set of categories while correcting the success expected probability. The values range from -1, which shows a complete disagreement between the categories and +1 showing a perfect agreement
(Comber et al., 2012).





### 2.4.5. - Classification scheme.

We used the R software to perform the ML, SVM, and RF classification process. R software packages are free and open-source (R: The R Project for Statistical Computing ). It offers a wide variety of functions for implementing algorithms and statistical analysis. The packages used were "e1071" (SVM implementation), "RandomForest" (RF implementation), "Raclass" (ML implementation) and "SP", "RASTER ", "RGDAL" (these 3 packages allow to read raster data in R). To train and validate the algorithms, ROI's were subset to provide 25% of the data for training and the remaining 75% for validation.

As mentioned in section 2.4.3., SVM and RF require some parameters defined by the user. In SVM classification, the first parameter is the kernel function type; here we selected the radial base Gaussian function (RBF), as being the most commonly used in remote sensing [49]. Two additional parameters "cost" (C parameter; which controls the cost paid by the SVM for erroneous classifications of a training point) and "gamma" (associated with the radial basis kernel function) needs to be defined, which in the first stage will have their values predetermined by the software. For the RF model, requires two parameters defined by the user, the number of variables available for each node division (mtry), and the number of decision trees (ntree) produced.

To obtain the highest classification accuracy, we perform an optimization parameter process to define the optimal learning algorithms. It is a necessary step as machines learning algorithms are sensitive to the parameters defined by the user (Mountrakis et al., 2011a). Its optimization is achieved by using the method of testing parameter combinations through cross-validation (10-fold) (Huang et al., 2002). The optimized parameters obtained were "gamma" (equal to 10) and "C" (equal to 1000) for SVM, and "ntree" (equal to 200) and "mtry" (equal to 3) for RF. Through the "epicalc" package, available in R, we create error matrices, with which it is possible to reach the global reliability value or total agreement of the models and the value of the kappa index.

### 2.4.6. - Classification

After the SVM and RF algorithms were trained, we validated the models by performing an accuracy assessment using the ROI's data set. The SVM and RF classification methods were applied for all corrected images. We only applied the ML method in the ASTER images, since it requires a longer time processing. In the end, SVM and RF are compared in Landsat, while the three SVM, RF and ML are compared with ASTER.

The results are raster maps or thematic maps that can be viewed using any geospatial software (QGIS). These thematic maps will control the six classes in which it has been desired to categorize the land cover (specified in table 2).

The colors of the thematic maps will have the same significances for all the images, where the black represents coverage of the city or exposed rock, the yellow for dry soil or with a little vegetal cover, the tan for humid grounds, the light green for low vegetation, dark green for tall vegetation and blue to represent water.

### 3. -Results

Table 3 shows the error matrices together with the values of overall agreement and kappa index for each of the models generated from different satellite images and different dates of acquisition. The land cover categories are represented by the numbers from 1 to 6.




**Table 3.** Error matrices, the total agreement and kappa index values, and user and producer accuracy for each of the models generated from the different satellite images and different acquisition dates. (1.-city; 2.-dry sediment; 3.-wet sediment; 4.-low vegetation; 5.-high vegetation; 6.- water).

| Satelite/model | | SVM | | | | | | RF | | | | | | MV | | | | | |
|---|---|---|---|---|---|---|---|---|---|---|---|---|---|---|---|---|---|---|---|
| | Class | 1 | 2 | 3 | 4 | 5 | 6 | 1 | 2 | 3 | 4 | 5 | 6 | | | | | | |
| Landsat | 1 | 562 | 16 | 0 | 16 | 5 | 0 | 552 | 21 | 3.5 | 3 | 7 | 0 | | | | | | |
| | 2 | 10 | 1965 | 8 | 25 | 26 | 3 | 21 | 1943 | 4 | 33 | 39 | 2 | | | | | | |
| | 3 | 3 | 34 | 2814 | 4 | 5 | 4 | 3 | 44 | 2799 | 4 | 4 | 2 | | | | | | |
| | 4 | 9 | 26 | 0 | 2742 | 111 | 0 | 10 | 26 | 1 | 2732 | 101 | 0 | | | | | | |
| | 5 | 11 | 22 | 5 | 148 | 3024 | 0 | 9 | 31 | 8 | 148 | 3028 | 0 | | | | | | |
| | 6 | 0 | 2 | 4 | 0 | 10 | 2672 | 0 | 1 | 4 | 0 | 1.5 | 2675 | | | | | | |
| | | Total agreement: 97.61% | | | Kappa : 0.965 | | | Total agreement : 97.46% | | | Kappa : 0.962 | | | | | | | | |
| | Class | 1 | 2 | 3 | 4 | 5 | 6 | 1 | 2 | 3 | 4 | 5 | 6 | 1 | 2 | 3 | 4 | 5 | 6 |
| ASTER | 1 | 968 | 19 | 2 | 9 | 4 | 4 | 965 | 23 | 1 | 17.5 | 2.5 | 5 | 775 | 20 | 1 | 0 | 43 | 999 |
| | 2 | 31 | 1325 | 64 | 109 | 39 | 0 | 27 | 1318 | 92 | 113 | 46 | 0 | 104 | 1291 | 240 | 328 | 94 | 0 |
| | 3 | 4 | 128 | 7512 | 6 | 16 | 43 | 5.5 | 117 | 7466 | 4 | 13 | 46 | 10 | 141 | 7312 | 5 | 17 | 293 |
| | 4 | 15 | 97 | 2 | 6489 | 290 | 0 | 21 | 99 | 4 | 6456 | 247 | 0 | 56 | 99 | 1 | 6128 | 410 | 0 |
| | 5 | 3 | 19 | 1 | 141 | 10074 | 0 | 5 | 32 | 4 | 163 | 10114 | 1 | 0 | 37 | 3 | 282 | 9634 | 215 |
| | 6 | 5 | 1 | 13 | 0 | 2 | 11367 | 4.5 | 1 | 28 | 0 | 3 | 11359 | 76 | 0 | 37 | 0 | 262 | 9906 |
| | | Total agreement: 96.93% | | | Kappa : 0.959 | | | Total agreement : 96.77 | | | Kappa : 0.958 | | | Total agreement : 89.45% | | | Kappa: 0.853 | | |
| User's accuracy (%) Landsat | | 93.8 | 96.4 | 98.2 | 95 | 94.2 | 99.4 | 94.1 | 95.1 | 98 | 95.2 | 93.9 | 99.7 | | | | | | |
| Producer accuracy(%)Landsat | | 94.5 | 95.2 | 99 | 93.4 | 95.1 | 99.7 | 92.8 | 94 | 99.3 | 93.5 | 95.2 | 99.8 | | | | | | |
| User's accuracy (%) ASTER | | 96.2 | 85 | 97.4 | 94.1 | 98.4 | 99.8 | 95.2 | 82.6 | 97.6 | 94.6 | 98 | 99.7 | 42 | 62 | 94 | 91.5 | 94.7 | 96.4 |
| Producer accuracy (%) ASTER | | 94.3 | 0.83 | 0.99 | 0.97 | 0.97 | 0.99 | 93.9 | 82.9 | 98.3 | 95.6 | 97 | 99.5 | 75.9 | 81.3 | 96.3 | 90.9 | 92.1 | 86.8 |

In addition, table 3 shows that the overall accuracies for the SVM and RF models, which vary between 96% and 97%, while the kappa index varies between 0.95 and 0.97. In contrast, for larger ensembles, overall accuracy is close to 98%. The average kappa index shows a value of 0.96 for both the SVM and RF algorithms, much higher than the 0.86 for the ML algorithm. The ML model has a lower precision (close to 90%) because it depends on a higher degree on the number of samples available for each category.

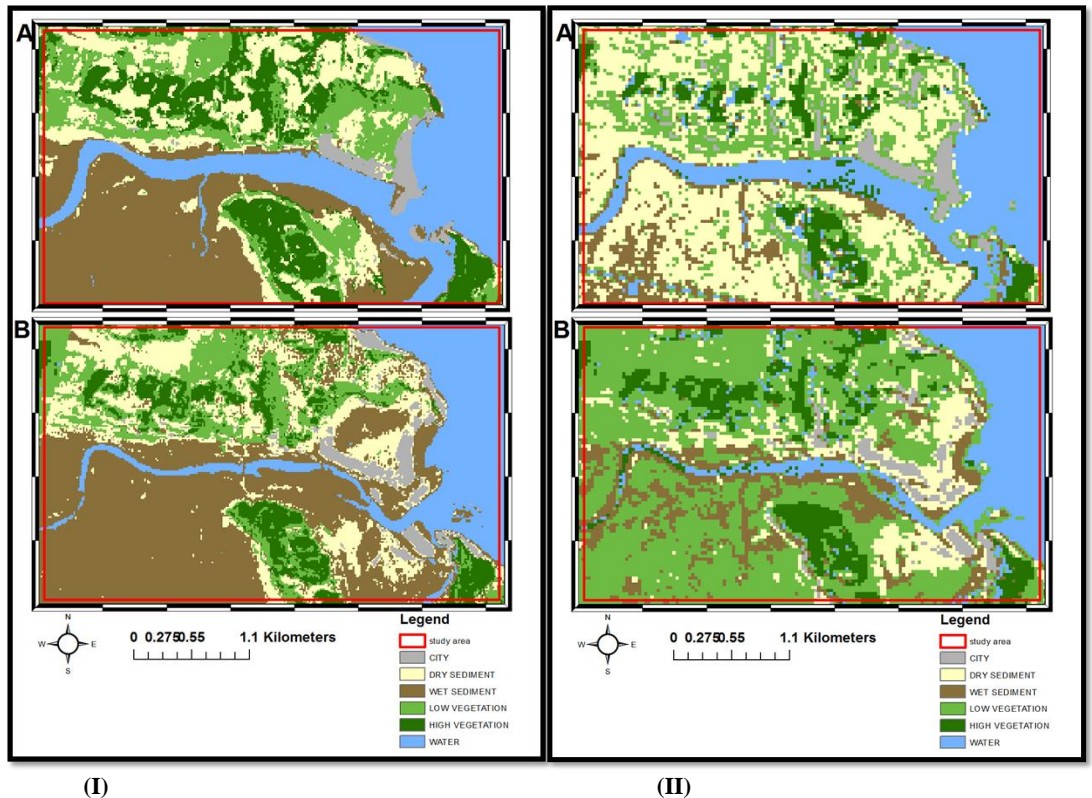

**(I)**    **(II)**

**Figure 3.** (I) Thematic maps for Tubul: A) Tubul thematic map, from ASTER image, for a period prior to 27 / F;
B) Tubul thematic map, from ASTER image, for a period after 27 / F. (II) Thematic maps for Tubul: A) Thematic
map Tubul, from Landsat image, for a period prior to 27 / F ; B) Thematic map of Tubul, from Landsat image, for
a period after 27 / F.

It should be noted that the accuracy levels are high in most cases (table 3). This is, related to the thematic
classes that used separate land coverings differentiable from each other; this helps in getting less
confusion between the classes. The number of training samples is abundant in all cases, helping to
optimize the algorithms. Thematic maps are presented in Figure 3.I and 3.II, resulting only for the SVM
method, since it was the one that showed the highest accuracy in performance, and changes in soil cover
will be described from these thematic maps

Table 4 present the soil cover description for Tubul, with a comparative approach before and after the
earthquake and tsunami. With this objective, for each category of the land cover, the numbers of pixels
classified are displayed. The percentages corresponding to the numbers of pixels selected for each coverage
regarding the total number of pixels in the image are also shown (%class). In the last column (Class
increase) the differences in percentage terms of the pixels selected for each category are presented,
comparing the two study periods. This provides us with value to discriminate if there was an increase or
decrease in each category (due to the earthquake and tsunami effects).





**Table 4.** Comparative table of changes in the thematic classes before and after the earthquake of 2010 for Tubul.

| | *Tubul* | | | | |
|---|---|---|---|---|---|
| | Before | | After | | |
| Class | Nº. of pixels | %Class | Nº. of pixels | % Class | Class increase |
| City | 1412 | 5.9% | 1817 | 7.6% | 1.7% |
| Dry Soil | 3123 | 13.0% | 4054 | 16.9% | 3.9% |
| Wet Floor | 7215 | 30.0% | 12093 | 50.3% | 20.3% |
| Vegetation Low | 4007 | 16.7% | 2561 | 10.6% | -6.1% |
| Vegetation High | 2922 | 12.1% | 1646 | 6.8% | -5.4% |
| Water | 5373 | 22.3% | 1884 | 7.8% | -14.5% |
| Total | 24055 | 100.0% | 24055 | 100.0% | 0% |


People reported that In Tubul a setback of several tens to hundred meters from the sea took place, because of the low slope of the area strongly affected by the co-seismic rise of $1.4 \pm 0.1$ meters (Quezada et al. 2012), forming a sandy beach (see Figure 3). After analyzing changes in the thematic map, the coastline receded approximately 200 meters. Another clear effect is the drying of the Tubul and Raqui rivers. The

thematic class of wet sediment replaced all these reversals of water cover, registering an increase of over 20% of this class, to the detriment of the thematic class of water and vegetation. Exposed rock coverings appeared, a product of landslides in steep areas and rock removal in coastal areas (see Figure 3).

The results presented in Table 4 describe the rate of change of land cover in percentage with respect to the total area covered in the study area. The cover corresponding to urban land increased from 5.9% to 7.6%,

dry land increased from 13% to 16.9%, wet soil had a large increase from 30% to 50.3%. The two categories related to vegetation had a decrease in the coverage area, the low vegetation from 16.7% to 10.6% and the high vegetation from 12.1% to 6.8%. Likewise, the aquatic cover decreased from 22.3% to 7.8%. All these variations in land cover give us a total change rate of 26.1%.

Among the land cover changes mentioned above the increase in the cover of wet soil, corresponding to the

appearance of wet soil in areas where water coverage existed before, due to the desiccation of the river basin and the retraction of the coastline, which exposed the seabed. The vegetation cover had important variations, especially the low vegetation cover, which corresponds to rocks and sediments landslides, and the entry of marine water into the interior areas, these processes causing vegetation death and subsequent soil exposure (without vegetation).

**4. 2 - Discussion**

For the image classification, the three algorithms show good results, although the SVM and RF learning algorithms have superior performance.

As far as the SVM classifier is concerned, from an algorithmic perspective, there is controversy about the kernel to be used and the selection and parameter optimization. Some authors postulate that optimization

processes do not contribute to an increase in classification accuracy (Zhang and Xie, 2013), while others show that the parameter fluctuations evince a great impact on precision. In this analysis, the applied parameter optimization process showed a high C parameter and a low gamma value, which agrees with what has been postulated elsewhere and particularly in (Maxwell et al., 2014). In addition, the accuracy of the algorithm will be subject to the choice of the kernel function (Huang et al., 2002); in this case, the

choice of the RBF kernel brought good results. All the above comes besides multiple works that show that there is empirical evidence to support the theoretical formulation and motivation behind SVM (Maxwell and Warner, 2015; Maxwell et al., 2014; Zhang and Xie, 2013).

For SVM, one of its most salient features is to generalize well from a limited amount and/or poor quality training data (Mountrakis et al., 2011a), as reflected in the high levels of overall reliability delivered


by models (see Table 3), although the representative pixels of the training samples for each class did not include only the category of pure soil coverage. This causes a deviation in the reflectivity that would correspond to only a ground cover, which implies a decrease in the training's quality data. The ability to get high accuracy of SVM, despite the described limitations, is in line with the concept of "support vector" that is based on only a few data points to define the hyperplane of the classifier, this process being

computationally lighter than other methods (Pal and Mather, 2005).

On the other hand, RF delivered very good results in the classification accuracy, showing a precision comparable to that of SVM. In addition, it is very easy to use, since it only has two parameters (the number of variables in the random subset of each node and the number of trees in the decision set) and is not very sensitive to their values (Rodriguez-Galiano et al., 2012). Regarding the number of decision trees, the

optimization reached an optimal value of 200, which agrees with other studies showing that as the number of trees increases it rises accuracy, but only to a certain range where accuracy stabilizes (Ghimire et al., 2012; Shi and Yang, 2019). For the number of random variables available within each node, the value obtained was 3. Though this value is low, it avoids the correlation between the trees (Breiman, 2001).

The RF algorithm has the advantage of generating an unbiased internal estimate of the generalization

error (OOB error) (Cánovas-García et al., 2017). This means that is not necessary to use an independent evaluation subset. However, in order to assess the classification accuracy as for the other algorithms, training and validation data were used to assess the classification accuracy in the same manner as for the other classification algorithms. Also, an evaluation set was used to measure the classification accuracy but randomness is desired in the evaluation set to avoid the bias generated in the measurement of the OOB

error. RF also provides an evaluation of variables importance (bands) for the general classification of the land cover categories and each category classification using the Ginni index and the OOB estimate (Breiman, 2001). In this study this estimate was not considered, since for the images analyzed, the number of bands was quite small (6 for Landsat and 3 for ASTER), so that reducing them would not generate a significant reduction in computational cost.

SVM and RF obtained similar values of global precision, classifying the land cover categories equally well, which is consistent with research that indicates a similar level of performance in terms of accuracy for both types of classifiers (Adam et al. 2014; Pal and Mather 2005). It should be noted that the high levels of accuracy obtained by learning algorithms, respond in part, by the high number of training samples used and the low number of bands that made up each image, as postulated in ( Pal and Foody, 2010).


**5. - Conclusion.**

The results presented here show that machine learning algorithms had an excellent performance in the classification of changes in land cover, facing a catastrophic process such as an earthquake with many aftershocks accompanied by a tsunami. These results respond to the good classification accuracy, the

optimal choice of parameters for the algorithms, thanks to implementing a parameter optimization. On the other hand, many training sample selections generate more robust algorithms. The results provide new perspectives on SVM and RF algorithm's performance in mapping's context of soil cover in large and heterogeneous areas. In addition, the results add to other research showing the superiority of learning algorithms versus more traditional methods, setting them as the best option for classifying land cover in

heterogeneous areas (Maxwell et al., 2018b; Melgani & Bruzzone, 2004).

Similar results were obtained to those shown in other studies with similar characteristics, we obtained comparable results, with the changes shown in the ground cover, a product of earthquake or tsunami effects analysis, through the implementation of image classification algorithms satellite (Ishihara et.al., 2017; Pandey et al.,2019).

This study gives a more local-scale approach to changes in the land cover before and after the 2010 event in Chile, capture changes in more limited areas, which stands out from other work done to analyze this event by generating thematic maps, which focused on a more regional scale (Rojas et al., 2013). In addition, results agree with those shown by other types of geographic and morphological analysis, which performed on-site cadasters for change quantification (Araya  and Carvajal 2016; Cienfuegos et al. 2014).

Those agree in the co-seismic uprising that produced a widening of beaches and rivers draining, and the sedimentary material entry or carried away by the tsunami, which both invaded vegetative areas or covers urban areas.



This work contributes to show that earthquakes and tsunamis, which are very rare, powerful and destructive natural events with deep consequences in landscape, could be quickly analyzed through passive satellite image and new machine learning methodologies, that can help to measure quickly and precisely not only the extent of a catastrophe but also its real effects on the territory. In addition, it can be established as a tool for the generation of risk maps for catastrophic events, serving as a risk management instrument both to improve territorial planning in coastal areas, optimize evacuation routes and artificial barriers creation to protect urban areas.

## 6. - References

Adam, E., Mutanga, O., Odindi, J., & Abdel-Rahman, E. M.: Land-use/cover classification in a heterogeneous coastal landscape using RapidEye imagery: evaluating the performance of random forest and support vector machines classifiers. *International Journal of Remote Sensing*, *35*(10), 3440–3458. https://doi.org/10.1080/01431161.2014.903435, 2014.

Ahmad, A., & Quegan, S.: Analysis of Maximum Likelihood Classification on Multispectral Data. In *Applied Mathematical Sciences* (Vol. 6), 2012.

Araya, C., & Carvajal, M.: Efectos geomorfológicos del tsunami de Chile de 2010 frente a la zona de máximo slip, revelados por imágenes satelitales y observaciones de campo: El caso del litoral arenoso La Trinchera, Región del Maule. *Investigaciones Geográficas*, (52), 5. https://doi.org/10.5354/0719-5370.2016.43260, 2016.

Belgiu, M., & Drăguţ, L.: Random forest in remote sensing: A review of applications and future directions. *ISPRS Journal of Photogrammetry and Remote Sensing*, *114*, 24–31. https://doi.org/10.1016/j.isprsjprs.2016.01.011, 2016.

Blanzieri, E., & Melgani, F.: Nearest neighbor classification of remote sensing images with the maximal margin principle. *IEEE Transactions on Geoscience and Remote Sensing*, *46*(6), 1804–1811. https://doi.org/10.1109/TGRS.2008.916090, 2008.

Bocco, M., Ovando, G., Sayago, S., & Willington, E.: Neural network models for land cover classification from satellite images. *Agricultura Tecnica*, *67*(4), 414–421. https://doi.org/10.4067/s0365-28072007000400009, 2007.

Breiman, L.: Random forests. *Machine Learning*, *45*(1), 5–32. https://doi.org/10.1023/A:1010933404324, 2001

Cánovas-García, F., Alonso-Sarría, F., Gomariz-Castillo, F., & Oñate-Valdivieso, F.: Modification of the random forest algorithm to avoid statistical dependence problems when classifying remote sensing imagery. *Computers & Geosciences*, *103*, 1–11. https://doi.org/10.1016/j.cageo.2017.02.012, 2017.

Chander, G., Markham, B. L., & Helder, D. L.: Summary of current radiometric calibration coefficients for Landsat MSS, TM, ETM+, and EO-1 ALI sensors. *Remote Sensing of Environment*, *113*(5), 893–903. https://doi.org/10.1016/j.rse.2009.01.007, 2009

Chuvieco, E.: *Teledtección ambiental : la observación de la Tierra desde el espacio*. Ariel, 2010.

Cienfuegos, R., Villagran, M., Aguilera, J. C., Catalán, P., Castelle, B., & Almar, R.: Video monitoring and field measurements of a rapidly evolving coastal system: the river mouth and sand spit of the Mataquito River in Chile. *Journal of Coastal Research*, *70*, 639–644. https://doi.org/10.2112/si70-108.1, 2014.

Comber, A., Fisher, P., Brunsdon, C., & Khmag, A.: Spatial analysis of remote sensing image classification accuracy. *Remote Sensing of Environment*, *127*, 237–246, 2012. https://doi.org/10.1016/j.rse.2012.09.005

Dhodhi, M. K., Saghri, J. A., Ahmad, I., & Ul-Mustafa, R.: D-ISODATA: A Distributed Algorithm for Unsupervised Classification of Remotely Sensed Data on Network of Workstations. *Journal of Parallel and Distributed Computing*, *59*(2), 280–301. https://doi.org/10.1006/jpdc.1999.1573, 1999.


Dye, M., Mutanga, O., & Ismail, R.: Examining the utility of random forest and AISA Eagle hyperspectral
          image data to predict Pinus patula age in KwaZulu-Natal, South Africa. *Geocarto International*,
          *26*(4), 275–289. https://doi.org/10.1080/10106049.2011.562308, 2011.

      Ghimire, B., Rogan, J., Galiano, V., Panday, P., & Neeti, N.: An evaluation of bagging, boosting, and
          random forests for land-cover classification in Cape Cod, Massachusetts, USA. *GIScience and
Remote Sensing*, *49*(5), 623–643. https://doi.org/10.2747/1548-1603.49.5.623, 2012.

      He, Y., Lee, E., & Warner, T. A.: A time series of annual land use and land cover maps of China from
          1982 to 2013 generated using AVHRR GIMMS NDVI3g data. *Remote Sensing of Environment*,
          *199*, 201–217. https://doi.org/10.1016/j.rse.2017.07.010, 2017.

      Huang, C., Davis, L. S., & Townshend, J. R. G.: An assessment of support vector machines for land cover
classification.    *International    Journal    of    Remote    Sensing*,    *23*(4),    725–749.
          https://doi.org/10.1080/01431160110040323, 2002.

      Ishihara, M., & Tadono, T.: Land cover changes induced by the great east Japan earthquake in 2011.
          *Scientific Reports*, *7*. https://doi.org/10.1038/srep45769, 2017.

      Kaiser, G., Burkhard, B., Römer, H., Sangkaew, S., Graterol, R., Haitook, T., … Sakuna-Schwartz, D.:
Mapping tsunami impacts on land cover and related ecosystem service supply in Phang Nga,
          Thailand.   *Natural   Hazards   and   Earth   System   Sciences*,   *13*(12),   3095–3111.
          https://doi.org/10.5194/nhess-13-3095-2013, 2013.

      Kavzoglu, T., & Colkesen, I.: A kernel functions analysis for support vector machines for land cover
          classification. *International Journal of Applied Earth Observation and Geoinformation*, *11*(5), 352–
359. https://doi.org/10.1016/j.jag.2009.06.002, 2009.

      Lawrence, R. L., & Moran, C. J.: The AmericaView classification methods accuracy comparison project:
          A rigorous approach for model selection. *Remote Sensing of Environment*, *170*, 115–120.
          https://doi.org/10.1016/j.rse.2015.09.008, 2015.

      Ma, Y., Chen, F., Liu, J., He, Y., Duan, J., & Li, X.: An Automatic Procedure for Early Disaster Change
Mapping    Based    on    Optical    Remote    Sensing.    *Remote    Sensing*,    *8*(4),    272.
          https://doi.org/10.3390/rs8040272, 2016.

      Martínez, C. D., Rojas, O., Edilia Jaque, D., Quezada, J., Daniela Vázquez, G., & Arturo Belmonte, I.:
          EFECTOS TERRITORIALES DEL TSUNAMI DEL 27 DE FEBRERO DE 2010 EN LA COSTA
          DE LA REGIÓN DEL BIO-BÍO, CHILE. In *Revista Geográfica de América Central Número
Especial EGAL,* 2011.

      Maxwell, A. E., & Warner, T. A.: Differentiating mine-reclaimed grasslands from spectrally similar land
          cover using terrain variables and object-based machine learning classification. *International Journal
          of Remote Sensing*, *36*(17), 4384–4410. https://doi.org/10.1080/01431161.2015.1083632, 2015.

      Maxwell, A. E., Warner, T. A., & Fang, F.: Implementation of machine-learning classification in remote
sensing: an applied review. *International Journal of Remote Sensing*, *39*(9), 2784–2817.
          https://doi.org/10.1080/01431161.2018.1433343, 2018ª.

      Maxwell, A. E., Warner, T. A., & Fang, F.: Implementation of machine-learning classification in remote
          sensing: An applied review. *International Journal of Remote Sensing*, Vol. 39, pp. 2784–2817.
          https://doi.org/10.1080/01431161.2018.1433343, 2018b.

Maxwell, A. E., Warner, T. A., Strager, M. P., & Pal, M.: *Combining RapidEye Satellite Imagery and Lidar
          for Mapping of Mining and Mine Reclamation*. https://doi.org/10.14358/PERS.80.2.179, 2014.

      Melgani, F., & Bruzzone, L.: Classification of hyperspectral remote sensing images with support vector
          machines. *IEEE Transactions on Geoscience and Remote Sensing*, *42*(8), 1778–1790.
          https://doi.org/10.1109/TGRS.2004.831865, 2004.

Moreira, E. P., & Valeriano, M.: Application and evaluation of topographic correction methods to improve
          land cover mapping using object-based classification. *International Journal of Applied Earth
          Observation and Geoinformation*, *32*(1), 208–217. https://doi.org/10.1016/j.jag.2014.04.006, 2014.





Mountrakis, G., Im, J., & Ogole, C.: ISPRS Journal of Photogrammetry and Remote Sensing Support vector machines in remote sensing: A review. *ISPRS Journal of Photogrammetry and Remote Sensing*, *66*, 247–259. https://doi.org/10.1016/j.isprsjprs.2010.11.001, 2011ª.

Mountrakis, G., Im, J., & Ogole, C.: ISPRS Journal of Photogrammetry and Remote Sensing Support vector machines in remote sensing: A review. *ISPRS Journal of Photogrammetry and Remote Sensing*, *66*, 247–259. https://doi.org/10.1016/j.isprsjprs.2010.11.001, 2011b.

Naidoo, L., Cho, M. A., Mathieu, R., & Asner, G.: Classification of savanna tree species, in the Greater Kruger National Park region, by integrating hyperspectral and LiDAR data in a Random Forest data mining environment. *ISPRS Journal of Photogrammetry and Remote Sensing*, *69*, 167–179. https://doi.org/10.1016/j.isprsjprs.2012.03.005, 2012

Pal, M, & Foody, G. M.: Feature selection for classification of hyperspectral data by SVM. In *IEEE Transactions on Geoscience and Remote Sensing* (Vol. 48), 2010.

Pal, Mahesh, & Mather, P. M. *An assessment of the effectiveness of decision tree methods for land cover classification*. https://doi.org/10.1016/S0034-4257(03)00132-9,2003

Pal, M. & Mather, P. M. Support vector machines for classification in remote sensing. *Int. J. Remote Sens.* **26**, 1007–1011, 2005.

Pandey, P. C., Koutsias, N., Petropoulos, G. P., Srivastava, P. K., & Ben Dor, E.: Land use/land cover in view of earth observation: data sources, input dimensions, and classifiers—a review of the state of the art. *Geocarto International*, 1–32. https://doi.org/10.1080/10106049.2019.1629647, 2019.

Quezada, J., Jaque, E., Fernández, A., Belmonte, A., & Martínez, C.: *Longitud de ruptura del terremoto Mw=8,8 del 27 de febrero de 2010 en el centro-sur de Chile*. https://doi.org/10.1007/s00024-011-0283-5, 2012.

R: The R Project for Statistical Computing.: from https://www.r-project.org/, 2019.

Rawat, J. S., & Kumar, M.: Monitoring land use/cover change using remote sensing and GIS techniques: A case study of Hawalbagh block, district Almora, Uttarakhand, India. *Egyptian Journal of Remote Sensing and Space Science*, *18*(1), 77–84. https://doi.org/10.1016/j.ejrs.2015.02.002, 2015.

Richmond, B., Szczuciński, W., Chagué-Goff, C., Goto, K., Sugawara, D., Witter, R., … Goff, J.: Erosion, deposition and landscape change on the Sendai coastal plain, Japan, resulting from the March 11, 2011 Tohoku-oki tsunami. *Sedimentary Geology*, *282*, 27–39. https://doi.org/10.1016/j.sedgeo.2012.08.005, 2012.

Rodriguez-Galiano, V. F., Ghimire, B., Rogan, J., Chica-Olmo, M., & Rigol-Sanchez, J. P.: An assessment of the effectiveness of a random forest classifier for land-cover classification. *ISPRS Journal of Photogrammetry and Remote Sensing*, *67*(1), 93–104. https://doi.org/10.1016/j.isprsjprs.2011.11.002, 2012.

Rogan, J., Franklin, J., Stow, D., Miller, J., Woodcock, C., & Roberts, D.: Mapping land-cover modifications over large areas: A comparison of machine learning algorithms. *Remote Sensing of Environment*, *112*(5), 2272–2283. https://doi.org/10.1016/j.rse.2007.10.004, 2008.

Rojas, C., Mauricio, V., Sergio, O., Peters, S., & Constanza, V.: Pre and post earthquake land use and land cover identification in Concepción. *Lecture Notes in Geoinformation and Cartography*, (199659), 223–231. https://doi.org/10.1007/978-3-642-32714-8_15, 2013.

Römer, H., Willroth, P., Kaiser, G., Vafeidis, A. T., Ludwig, R., Sterr, H., & Revilla Diez, J.: Potential of remote sensing techniques for tsunami hazard and vulnerability analysis-a case study from Phang-Nga province, Thailand. *Natural Hazards and Earth System Science*, *12*(6), 2103–2126. https://doi.org/10.5194/nhess-12-2103-2012, 2012.

Sarun, S., Vineetha, P., Kumar, R., & Jayalekshmi, V.: Spatial Analysis of Land Use and Land Cover Changes Using Spectral Indices in the Tsunami Affected Areas in Kerala, India. *Journal of Geography, Environment and Earth Science International*, *15*(4), 1–11. https://doi.org/10.9734/jgeesi/2018/41927, 2018


Satheesh Kumar, C., Arul Murugan, P., Krishnamurthy, R. R., Prabhu Doss Batvari, B., Ramanamurthy, M. V., Usha, T., & Pari, Y.: Inundation mapping - A study based on December 2004 Tsunami Hazard along Chennai coast, Southeast India. *Natural Hazards and Earth System Science*, *8*(4), 617–626. https://doi.org/10.5194/nhess-8-617-2008, 2008.

Shi, D., & Yang, X.: *An Assessment of Algorithmic Parameters Affecting Image Classification Accuracy by Random Forests*. https://doi.org/10.14358/PERS.82.6.407, 2019.

Soenen, S. A., Peddle, D. R., & Coburn, C. A.: SCS+C: A Modified Sun-Canopy-Sensor Topographic Correction in Forested Terrain. *IEEE TRANSACTIONS ON GEOSCIENCE AND REMOTE SENSING*, *43*(9). https://doi.org/10.1109/TGRS.2005.852480, 2005.

Soto, M. V., Arriagada, J., Castro-Correa, C. P., Ibarra, I., & Rodolfi, G.: Condiciones geodinámicas derivadas del terremoto y tsunami de 2010 en la costa de Chile central. El caso de Pichilemu. *Revista de Geografía Norte Grande*, *60*, 79–95. https://doi.org/10.4067/s0718-34022015000100005, 2015.

Tappin, D. R., Evans, H. M., Jordan, C. J., Richmond, B., Sugawara, D., & Goto, K.: Coastal changes in the Sendai area from the impact of the 2011 Tōhoku-oki tsunami: Interpretations of time series
satellite images, helicopter-borne video footage and field observations. *Sedimentary Geology*, *282*, 151–174. https://doi.org/10.1016/j.sedgeo.2012.09.011, 2012.

Tralli, D. M., Blom, R. G., Zlotnicki, V., Donnellan, A., & Evans, D. L.: Satellite remote sensing of earthquake, volcano, flood, landslide and coastal inundation hazards. *ISPRS Journal of Photogrammetry and Remote Sensing*, *59*(4), 185–198.
https://doi.org/10.1016/j.isprsjprs.2005.02.002, 2005.

Vapnik, V. N.: *The nature of statistical learning theory*. Springer, 1995.

Wu, J., Wang, T., Pan, K., Li, W., & Huang, X.: Assessment of forest damage caused by an ice storm using multi-temporal remote-sensing images: a case study from Guangdong Province. *International Journal of Remote Sensing*, *37*(13), 3125–3142. https://doi.org/10.1080/01431161.2016.1194544,
2016.

Yonezawa, C.: Maximum likelihood classification combined with spectral angle mapper algorithm for high resolution satellite imagery. *International Journal of Remote Sensing*, *28*(16), 3729–3737. https://doi.org/10.1080/01431160701373713, 2007.

Zhang, C., & Xie, Z.: Object-based vegetation mapping in the kissimmee river watershed using hymap data
and machine learning techniques. *Wetlands*, *33*(2), 233–244. https://doi.org/10.1007/s13157-012-0373-x, 2013.
