# Peer review of "Comparison of machine learning classification algorithms for land cover change in a coastal area affected by the 2010 Earthquake and Tsunami in Chile."

_Natural Hazards and Earth System Sciences, 2020_

## Referee Comment (RC1) · Anonymous Referee #1 · 24 Apr 2020

In this work, the authors implement two machine learning classification methods on optical satellite images of Landsat and ASTER and compare the accuracy of the results. They have applied the Maximum Likelihood method on the ASTER as well and compared its result with the machine learning-based methods. The document is well structured and written. However, the novelty of the work is limited or unclear.

In addition to this general comment, please find below some more specific comments:

Page 8, Figure 3: 1- I do not understand the reason for using both Landsat and ASTER

images? Does it add any further information?

2- Considering that the time difference between the Landsat and ASTER images is one-two weeks, please explain the reason for the differences between the thematic maps (I-A with II-A and I-B with II-B).

3- As the title of the manuscript is a comparison of the machine learning methods, I suggest adding the thematic maps created by applying the RF method. And compare the results in more detail.

4- Please add some signs or vectors in the map, so that the interpretation provided in page 9 becomes more understandable.

Page 4, paragraph 105: Please explain the 'correction C' briefly. Page 4, figure 2: What does ND stand for? 'mtree' should be 'ntree'? Page 6, paragraph 165, last line: Possible typo: Is it possible that the authors meant "75% of the data for training and the remaining 25% for validation", instead of 25% of the data for training and the remaining 75% for validation"? Page 6, paragraph 170, line 3: "[49]" Please use a unique style for citation. Page 7, Table 3: 1- typo: "MV" instead of "ML". 2- It seems that the values in the last row, second to sixth columns should be multiplied by 100?

Looking forward to a revised version of the paper.

---

## Referee Comment (RC2) · Anonymous Referee #2 · 28 Apr 2020

This paper evaluates the performance of a few basic classification approaches such as random forest and support vector machines to assess changes caused by the earthquake and tsunami that occurred on Feb 27, 2010, concentrating in the Tubul town area, in central Chile.

From the theoretical perspective, with my full respect to the work of the authors, this paper does not add something new compared to the existing literature. In more detail, this work only investigates the performance of three basic pixel-wise classifiers, i.e., ML, SVM, and RF, and the conclusions made out of the comparison are already known
for years. The authors could investigate many more classifiers to make more comprehensive conclusions. Obtaining very similar classification results for both RF and SVM reveals the fact that too many training samples are used for the classification task. Maybe I missed something, but since the training samples were chosen randomly, it is necessary to iterate the classifiers at least 10 times and report the mean and standard deviation values to avoid any biases induced in the training data. In addition, the effect of the number of training samples on the classification results could also be evaluated. The comparison of processing time consumed by different approaches is also necessary.

There are a number of format inconsistencies through spacing and indenting. There are also several grammatical mistakes and typos. What do you mean by MV in Table 3? Do you mean ML?

---

## Referee Comment (RC3) · Anonymous Referee #3 · 10 May 2020

The authors have investigated the performance of three different classifiers, namely Random Forest (RF), Support Vector Machine (SVM), and Maximum Likelihood classifier (MLC), on the detection of changes, which occurred by a natural hazard. The research question in this manuscript is not much meaningful. This is because detecting changes using comparing the results of classification in two dates (before and after the event) is not the most accurate and time-efficient approach, which are of great importance for natural disaster management. In addition, there is no need to compare two non-parametric approaches (RF and SVM) with a parametric MLC when the

answer is already known. Even for comparing the performance of different methods for a classification problem, I would suggest comparing the performance of RF and SVM with the more recently proposed approaches, including XGBosst and CNN algorithms. Furthermore, the paper is not well written and there is not enough novelty in this work. The extensive editing of the English language and style are also required for this manuscript.

---

## Author Comment (AC1) · 19 May 2020

we appreciate your comments. 10 iterations were performed in relation to the training samples, but we did not report it in the manuscript. Now we will repeat the sampling process but we will evaluate the effects of the training sample size on the results of the classification algorithms. 10 subsets of training data will be created by increasing the number of pixels per class in each subset. New pixel-wise classifiers algorithms will also be implemented (DT,KNN, ANN, DNN).

---

## Author Comment (AC2) · 19 May 2020

We understand your comments. We will incorporate more recent pixel-wise classifiers algorithms and compare the performance for different sizes of training samples. Thanks for your comments

---

## Author Comment (AC3) · 2 Jul 2020

In our opinion and to our knowledge to date (02-07-20), in the field of tsunami effects, this study represents a first evaluation of emerging machine learning algorithms for land cover classification using multispectral data on tsunami-affected areas. In that aspect the work is a contribution and should serve from now on as a reference and basis of work for possible studies related to the development of risk maps and damage cadastres. In turn, if we look at the literature, we do not see much work on data mining methodologies applied to remote sensing for earthquakes. So it could also serve as a

reference.

It is true, many of these self-learning algorithms have been widely used for remote sensing, especially after the explosion of data mining and notably in the case of landslides which is a nearby field of research, but it does not involve such expensive surface areas as ours (Wang et al. (2020), A. Merghadi et al. (2020)

However, and here we thank the reviewers, we now apply the Extreme Gradient Boosting (XGB) and Deep Neural Network (DNN) algorithms that have been used mostly in data science and have only recently been shown to work in remote sensing. Also, although comparisons between these types of classifiers can be found in the existing literature, they generally do not provide the space needed to find the best combination of parameters, analysis with different sizes and quality of training samples. These caries are incorporated in the new version of the manuscript.

Finally, our basic objective was to detect a fast and robust methodology that would easily account for a first approximation of visible changes after a tsunami and eventually an earthquake. In this sense, the results of this work allow us to affirm that the less complex methodologies obtain the same results as the more complex ones. Therefore, not knowing even if the results can be generalized (here is a study theme for application in more areas), it seems to us that we also achieved that contribution.

---

## Author Comment (AC4) · 2 Jul 2020

The two classes of images were incorporated to refer to the accuracy of the classifiers in images with different resolution. It is important for us to see how much better resolution is provided, and whether or not this is significant in the results of change and locally where. That is to say, how is its performance with lower resolution pixels that have a more heterogeneous variability within each pixel compared to higher resolution images, where you would have a more homogeneous reality within the pixel. This gives us a measure of how different algorithms respond to different qualities of training

samples.

It is important to recognize if there are large changes in the accuracy of the classification by increasing the resolution of the data, because one may be using the results of higher resolution satellite missions but sometimes the simplest thing is the best, especially when our goal is to create algorithms that are perennial and easy to use. ASTER data have better resolution than LANDSAT data, but the former will be discontinued. We could also have used other databases, with even lower resolution (modis), but we think that comparing those two databases is enough.

---

## Author Comment (AC5) · 2 Jul 2020

The thematic maps shown in I-A, II-A, I-B and II-B are now represented in thematic maps III, IV, I and II in Figure 4, respectively. Since the intention of this research is not to characterize the vegetation present in the total image, we decided to limit the area of classification, in relation to the maps presented in the manuscript, to a zone closer to the area of interest (see thematic maps in figure 4). Considering the differences present in the corresponding images at different dates, it should be considered that the region corresponding to the images is approximately 36°S and has a Mediterranean climate

where the summer seasons can be very dry or show sporadic rains that cause notable changes in the vegetation present. The Landsat and Aster images used training data that did not necessarily include the same pixels. Each of these pixels do not represent equally the homogeneity of the area they characterize, since the spatial resolution of the pixel is different between both satellites (30mx30m for landsat and 15mx15m for ASTER ), therefore the thematic maps resulting from the classification processes for both images should not necessarily be the same. Also, to certify the reliability of the results, new training zone samples were made based on high resolution google earth images. The selection of representative pixels for each of the classes was restricted to a maximum of 2000 samples with the highest possible land cover hogenity. This process brought greater visual similarity between the Aster and Landsat image theme maps.
* * *
[Figure]

**Figure 4.** The thematic maps generated by the SVM classification algorithm focused on Tubul for the pre- and post-tsunami period (27-F). (I) & (II): Thematic maps for Landsat image. (III) & (IV): Thematic maps for Aster image.

**Fig. 1.** The thematic maps generated by the SVM classification algorithm focused on Tubul for the pre- and post-tsunami period (27-F).

---

## Author Comment (AC6) · 2 Jul 2020

The thematic maps for all the ML algorithms are presented in the appendix. The ML algorithms produced similar thematic maps, coherent in the classes and with a minimum of spots. By using McNemar's chi-squared ( $\chi$ 2) test we compared the significance of the differences between the results of the error matrices of the algorithms, showing p-values lower than 5624e-7, so there are no significant differences between the results shown by the different thematic maps (thematic maps in Figure 1 to 4). For these reasons, the analysis of land cover changes is performed based on the best performance

algorithm. In figure 5 and 6 are presented a tables with the results of Average overall accuracies and their coefficient of variation for ML algorithms applied in the 10 training subsamples. In figure 6 are summarized the Wilcoxon test results between different model to evaluate the significance of the differences in accuracy of the algorithms. In figure 7 a comparison between The highest average value obtained of overall accuracy for the ML classifiers is presented.
Appendix B

Fig B1. The thematic maps generated by the NB, KNN, GBM, DNN, MARS, RF, SVM, XGB classifications algorithms, focused on Tubul before 27-F. Thematic maps for Landsat image.

**Fig. 1.** The thematic maps generated by the NB, KNN, GBM, DNN, MARS, RF, SVM, XGB classifications algorithms, focused on Tubul before 27-F. Thematic maps for Landsat image.

---

## Author Comment (AC7) · 2 Jul 2020

Thank you for the comment that allows us to improve the representation. Now, in the thematic maps (see figure 5), we defined polygons that shows areas where representative changes in land cover classes were observed. The polygons are used to help understand the values of changes in classes represented in manuscript

**Fig. 1.** The thematic maps generated by the SVM classification algorithm focused on Tubul for the pre- and post-tsunami period (27-F). (I) & (II): Thematic maps for Landsat image. (III) & (IV): Thematic maps

---

## Author Comment (AC8) · 3 Jul 2020

Thanking you for your comment, which prompted us to investigate the performance of more classifiers. In a new version of manuscript, we investigate 6 new algoithms, over a total of 8 algorithms K-Nearest Neighbor (KNN), Multivariate Adaptative Regression Spline (MARS), Gradient Boosting Machine (GBM), Support Vector Machine (SVM), Random Forest (RF), Extreme Gradient Boosting (XGB), Deep Neural Network (DNN), and one parametric algorithm Naïve Bayes (a Maximum Likelihood variant).

[Figure]

In order to compare the precision differences of the non-parametric ML algorithms with the classic parametric methodologies such as Maximum Likelihood, the NB algorithm was selected. The algorithms were chosen on reviewers recommendation, as is the case with Extreme Gradient Boosting (XGB) and Deep Neural Network (DNN), but also selected in consideration of the literature and the success they had had in various remote sensing experiences. Gradiente boosting machine (GBM) was chosen, mainly because belongs to a family of boosting algorithms as well as XGB. Therefore, we will have a performance comparison between a classic boosting machine model and a new implementation of it. SVM, RF and KNN were selected because there are important results that show their reliability for the classification of satellite images, so they are a good comparative scale in accuracy for the new families of classification algorithms. MARS was one of the most successful algorithms in previous decades. However aside the historical point it has a shorter kernel, but more adjusted than the others, that's why we thought it would be interesting to attach it.

---

## Author Comment (AC9) · 3 Jul 2020

Thank for this comment which allowed to improve the statistical significance of the work. For the process to achieve statistical confidence, classifiers are now iterated 100 times in 10 separate sub-sample data sets. We report the mean values and standard deviation in the manuscript (see section 3.- result) for each of the 10 training sample subgroups (see tables en figure 1 and 2). To evaluate the effect of training sample sizes as well as the performance of the classification algorithms on classification accuracies, we randomly divided the training sample data into 10 different data sets. The procedure

described in section 2.4.5. - Classification scheme" of the manuscript. These data sets correspond to 10%, 20%, 30%, 40%, 50%, 60%, 70%, 80%, 90%, 100% of the total training data set. We made sure that each subset had the same proportions of training samples per soil cover class as the original set. To ensure that this occurs, a technique called "stratified holdout sampling" is used. This technique is implemented in the CreateDataPartition() function of the caret package. A fixed validation sample is reserved for the performance evaluation of all classifiers, corresponding to 30% of the initial total of the training samples. This sample is obtained in the same way as the training samples. However, it is reserved only for the validation of the algorithms trained in the different sizes of training sub-samples. Also applied the Wilcoxon signed-rank test to evaluate and assess the statistical significance of systematic pairwise differences between the ML models at the significant level $\alpha$ = 5%. The results of the p-values of the significance test are presented in the table in figure 3 below.

———————————————

| Algorithm | Landsat | | | | | | | | | | | | | | | | | | |
|---|---|---|---|---|---|---|---|---|---|---|---|---|---|---|---|---|---|---|---|
| | 10% | | 20% | | 30% | | 40% | | 50% | | 60% | | 70% | | 80% | | 90% | | 100% | |
| | mean | stde | mean | stde | mean | stde | mean | stde | mean | stde | mean | stde | mean | stde | mean | stde | mean | stde | mean | stde |
| NB | 0.895 | 0.013 | 0.908 | 0.011 | 0.909 | 0.010 | 0.912 | 0.008 | 0.913 | 0.006 | 0.914 | 0.005 | 0.915 | 0.004 | 0.914 | 0.003 | 0.915 | 0.002 | 0.914 | 0.002 |
| KNN | 0.911 | 0.009 | 0.922 | 0.008 | 0.927 | 0.007 | 0.931 | 0.006 | 0.934 | 0.005 | 0.937 | 0.005 | 0.938 | 0.004 | 0.938 | 0.003 | 0.939 | 0.001 | 0.939 | 0.001 |
| MARS | 0.884 | 0.014 | 0.904 | 0.010 | 0.911 | 0.007 | 0.918 | 0.007 | 0.921 | 0.005 | 0.924 | 0.005 | 0.925 | 0.004 | 0.925 | 0.003 | 0.926 | 0.001 | 0.927 | 0.001 |
| GBM | 0.899 | 0.012 | 0.917 | 0.010 | 0.925 | 0.008 | 0.933 | 0.007 | 0.936 | 0.005 | 0.940 | 0.004 | 0.942 | 0.003 | 0.943 | 0.004 | 0.945 | 0.003 | 0.946 | 0.002 |
| SVM | 0.916 | 0.009 | 0.932 | 0.008 | 0.939 | 0.006 | 0.943 | 0.005 | 0.947 | 0.004 | 0.950 | 0.004 | 0.952 | 0.003 | 0.953 | 0.002 | 0.955 | 0.001 | 0.955 | 0.001 |
| RF | 0.917 | 0.010 | 0.927 | 0.007 | 0.933 | 0.006 | 0.936 | 0.006 | 0.939 | 0.005 | 0.940 | 0.005 | 0.941 | 0.004 | 0.943 | 0.004 | 0.943 | 0.003 | 0.942 | 0.002 |
| DNN | 0.925 | 0.011 | 0.939 | 0.009 | 0.934 | 0.009 | 0.934 | 0.008 | 0.938 | 0.008 | 0.945 | 0.007 | 0.946 | 0.005 | 0.948 | 0.004 | 0.949 | 0.001 | 0.949 | 0.001 |
| XGB | 0.904 | 0.016 | 0.920 | 0.012 | 0.927 | 0.010 | 0.933 | 0.007 | 0.939 | 0.006 | 0.942 | 0.005 | 0.946 | 0.004 | 0.947 | 0.003 | 0.951 | 0.001 | 0.952 | 0.001 |

**Fig. 1.** Table 4. Average overall accuracies and their coefficient of variation for ML algorithms applied in all training sample size (Landsat images).

| Aster | | | | | | | | | | | | | | | | | | | |
|---|---|---|---|---|---|---|---|---|---|---|---|---|---|---|---|---|---|---|---|
| Algorithm | 10% | | 20% | | 30% | | 40% | | 50% | | 60% | | 70% | | 80% | | 90% | | 100% | |
| | mean | stde | mean | stde | mean | stde | mean | stde | mean | stde | mean | stde | mean | stde | mean | stde | mean | stde | mean | stde |
| NB | 0.924 | 0.013 | 0.923 | 0.007 | 0.931 | 0.007 | 0.934 | 0.011 | 0.932 | 0.007 | 0.933 | 0.005 | 0.933 | 0.005 | 0.932 | 0.002 | 0.934 | 0.001 | 0.934 | 0.001 |
| KNN | 0.925 | 0.014 | 0.940 | 0.010 | 0.954 | 0.006 | 0.957 | 0.005 | 0.964 | 0.004 | 0.966 | 0.002 | 0.967 | 0.003 | 0.970 | 0.002 | 0.971 | 0.002 | 0.971 | 0.002 |
| MARS | 0.942 | 0.010 | 0.952 | 0.021 | 0.949 | 0.011 | 0.959 | 0.022 | 0.959 | 0.021 | 0.967 | 0.004 | 0.967 | 0.007 | 0.969 | 0.014 | 0.969 | 0.002 | 0.970 | 0.002 |
| GBM | 0.935 | 0.010 | 0.958 | 0.008 | 0.965 | 0.004 | 0.968 | 0.004 | 0.970 | 0.002 | 0.973 | 0.003 | 0.974 | 0.002 | 0.975 | 0.002 | 0.975 | 0.001 | 0.975 | 0.001 |
| SVM | 0.944 | 0.012 | 0.962 | 0.008 | 0.967 | 0.005 | 0.971 | 0.006 | 0.975 | 0.003 | 0.976 | 0.004 | 0.976 | 0.003 | 0.977 | 0.001 | 0.979 | 0.001 | 0.979 | 0.001 |
| RF | 0.930 | 0.015 | 0.957 | 0.014 | 0.963 | 0.003 | 0.965 | 0.004 | 0.967 | 0.003 | 0.970 | 0.002 | 0.971 | 0.002 | 0.972 | 0.002 | 0.974 | 0.001 | 0.973 | 0.001 |
| DNN | 0.945 | 0.002 | 0.951 | 0.001 | 0.961 | 0.002 | 0.964 | 0.001 | 0.967 | 0.002 | 0.968 | 0.004 | 0.969 | 0.003 | 0.972 | 0.001 | 0.973 | 0.001 | 0.973 | 0.001 |
| XGB | 0.935 | 0.013 | 0.957 | 0.009 | 0.965 | 0.004 | 0.966 | 0.005 | 0.969 | 0.002 | 0.971 | 0.004 | 0.974 | 0.002 | 0.973 | 0.002 | 0.974 | 0.001 | 0.976 | 0.001 |

**Fig. 2.** Table 5. Average overall accuracies and their coefficient of variation for ML algorithms applied in all training sample size (Aster images).

[Figure]

| Landsat | | | | Aster | | | | Landsat-Aster | |
|---|---|---|---|---|---|---|---|---|---|
| No | p.value | No | p.value | No | p.value | No | p.value | No | p.value |
| NB vs. KNN | <0.001 | MARS vs. SVM | <0.001 | NB vs. KNN | <0.001 | MARS vs. SVM | <0.001 | NB | <0.001 |
| NB vs. MARS | <0.001 | MARS vs. RF | <0.001 | NB vs. MARS | <0.001 | MARS vs. RF | 0.064 | KNN | <0.001 |
| NB vs. GBM | <0.001 | MARS vs. DNN | 0.084 | NB vs. GBM | <0.001 | MARS vs. DNN | 0.004 | MARS | <0.001 |
| NB vs. SVM | <0.001 | MARS vs. XGB | 0.033 | NB vs. SVM | <0.001 | MARS vs. XGB | 0.037 | GBM | 0.006 |
| NB vs. RF | <0.001 | GBM vs. SVM | <0.001 | NB vs. RF | <0.001 | GBM vs. SVM | <0.001 | SVM | 0.006 |
| NB vs. DNN | <0.001 | GBM vs. RF | <0.001 | NB vs. DNN | <0.001 | GBM vs. RF | <0.001 | RF | <0.001 |
| NB vs. XGB | <0.001 | GBM vs. DNN | 0.084 | NB vs. XGB | <0.001 | GBM vs. DNN | 0.084 | DNN | <0.001 |
| KNN vs. MARS | 0.906 | GBM vs. XGB | 0.033 | KNN vs. MARS | 0.906 | GBM vs. XGB | 0.033 | XGB | <0.001 |
| KNN vs. GBM | <0.001 | SVM vs RF | 0.006 | KNN vs. GBM | <0.001 | SVM vs RF | 0.006 | | |
| KNN vs. SVM | <0.001 | SVM vs DNN | 0.004 | KNN vs. SVM | <0.001 | SVM vs DNN | 0.004 | | |
| KNN vs. RF | <0.001 | SVM vs XGB | 0.006 | KNN vs. RF | <0.001 | SVM vs XGB | 0.006 | | |
| KNN vs. DNN | 0.006 | RF vs XGB | 0.232 | KNN vs. DNN | 0.006 | RF vs XGB | 0.023 | | |
| KNN vs. XGB | <0.001 | RF vs DNN | 0.020 | KNN vs. XGB | <0.001 | RF vs DNN | 0.088 | | |
| MARS vs. GBM | 0.041 | DNN vs XGB | 0.084 | MARS vs. GBM | 0.041 | DNN vs XGB | 0.084 | | |

**Fig. 3.** Table 6. Wilcoxon test results between different models for each image type and between images (significance achieved at $p < 0.05$).

---

## Author Comment (AC10) · 3 Jul 2020

we appreciate your feedback. In our opinion and to the best of our knowledge to date (02-07-20), in the field of tsunami effects, this study represents the first evaluation of emerging machine learning algorithms for land cover classification using multispectral data on tsunami-affected areas. In that aspect, the work is a contribution. It could serve from now on as a reference and basis of work for possible studies related to the development of risk maps and damage cadastres. In turn, if we look at the literature, we do not see much work either on machine learning methodologies applied to remote

sensing for earthquakes hazard zones. So it could also serve as a reference.

In addition, our primary objective was to detect a rapid and robust methodology that would easily account for the first approximation of visible changes after a tsunami and, eventually, an earthquake. In this sense, the results of this work allow us to affirm that the less complicated methodologies obtain the same results as the more complex ones.

Therefore, not knowing at this stage if the results can be generalized (here is a study theme for application of ML comparison in more areas), it seems to us that we also achieved that contribution.

---

## Author Comment (AC11) · 3 Jul 2020

we are very grateful for the referee recommendations. It allowed us to learn and test new methodologies. We were quite curious about implementing Extreme Gradient Boosting (XGB, it has received much attention lately) and Deep Neural Network (DNN). Both have been used mostly in data science and only recently proven to work in remote sensing.

In total, the new version of the manuscript, we investigate 6 new algorithms, over a total

of 8 algorithms. K-Nearest Neighbor (KNN), Multivariate Adaptative Regression Spline (MARS), Gradient Boosting Machine (GBM), Support Vector Machine (SVM), Random Forest (RF), Extreme Gradient Boosting (XGB), Deep Neural Network (DNN), and one parametric algorithm Naïve Bayes (a Maximum Likelihood variant). Gradient boosting machine (GBM) was chosen because it belongs to a boosting algorithms family of as well as XGB. Therefore, we could also investigate the performance comparison between the classic boosting machine model and its new implementation. We selected others in consideration of the literature and the success they had had in various remote sensing experiences. Thus, the SVM, RF, and KNN methods were selected after outstanding results showing their reliability for satellite image classification. MARS was one of the most successful (and mysterious) algorithms in previous decades. To compare the precision differences of the non-parametric ML algorithms with the standard parametric methodologies such as the Maximum Likelihood, we selected the NB algorithm.